# A Theoretical Study of Fe Adsorbed on Pure and Nonmetal (N, F, P, S, Cl)-Doped Ti_3_C_2_O_2_ for Electrocatalytic Nitrogen Reduction

**DOI:** 10.3390/nano12071081

**Published:** 2022-03-25

**Authors:** Heng Luo, Xiaoxu Wang, Chubin Wan, Lu Xie, Minhui Song, Ping Qian

**Affiliations:** 1Department of Physics, University of Science and Technology Beijing, Beijing 100083, China; s20190798@xs.ustb.edu.cn (H.L.); qianping@ustb.edu.cn (P.Q.); 2Beijing Advanced Innovation Center for Materials Genome Engineering, University of Science and Technology Beijing, Beijing 100083, China; b20170555@xs.ustb.edu.cn (L.X.); s20191385@xs.ustb.edu.cn (M.S.); 3DP Technology, Beijing 100083, China; wangxx@dp.tech

**Keywords:** DFT, MXene, nitrogen reduction, electrocatalysis, Gibbs free energy

## Abstract

The possibility of using transition metal (TM)/MXene as a catalyst for the nitrogen reduction reaction (NRR) was studied by density functional theory, in which TM is an Fe atom, and MXene is pure Ti_3_C_2_O_2_ or Ti_3_C_2_O_2−*x*_ doped with N/F/P/S/Cl. The adsorption energy and Gibbs free energy were calculated to describe the limiting potentials of N_2_ activation and reduction, respectively. N_2_ activation was spontaneous, and the reduction potential-limiting step may be the hydrogenation of N_2_ to *NNH and the desorption of *NH_3_ to NH_3_. The charge transfer of the adsorbed Fe atoms to N_2_ molecules weakened the interaction of N≡N, which indicates that Fe/MXene is a potential catalytic material for the NRR. In particular, doping with nonmetals F and S reduced the limiting potential of the two potential-limiting steps in the reduction reaction, compared with the undoped pure structure. Thus, Fe/MXenes doped with these nonmetals are the best candidates among these structures.

## 1. Introduction

Ammonia is a raw material for the production of various fertilizers and is a potential energy source that is easy to store and transport, environmentally friendly, and relatively safe. Ammonia synthesis is important in agricultural production and energy development. However, most ammonia synthesis still relies on the Hubble–Bosch method proposed in the 20th century, which requires harsh reaction conditions (400–600 °C and 20–40 MPa) [1,2,3]. This method consumes a large amount of energy and causes significant greenhouse gas emissions [4]. In addition, other negative effects, such as adverse effects on the equipment under high-temperature and high-pressure conditions, need to be considered. Therefore, the development of environmentally friendly and less energy-demanding methodologies for NH_3_ synthesis is urgently needed. Electrocatalytic ammonia synthesis has attracted increasing attention owing to its high efficiency and environmental friendliness. The introduction of electrical energy has a remarkable influence on N_2_ activation and changes the reaction pathways [5], which is beneficial for the development of new stable and efficient catalysts.

New catalysts can be developed from unique structures, such as core–shell Ni–Au nanoparticles for CO_2_ hydrogenation [6], or from new materials. The excellent physical, electronic, and chemical properties of two-dimensional (2D) materials have attracted extensive scientific research [7,8,9,10,11,12,13]. In addition, 2D materials, such as molybdenum disulfide, graphene, and metal–organic frameworks (MOFs) [14,15,16], have emerged as potential candidates for electrochemical nitrogen reduction reactions (NRRs). Notably, MXene, a new member of the 2D material family that joined in 2011 [17], has developed rapidly in the past nine years [17,18,19]. The general formula of MXene is M*_n_*_+1_X*_n_*T*_x_*, where M represents early transition metals (TMs), X represents carbon or nitrogen, T*_x_* represents the surface functional groups O, OH, or F, and *n* = 1, 2, 3. MXenes are synthesized by the chemical etching of A layers in the MAX (M*_n_*_+1_AX*_n_*) phase. Although a variety of 2D MXenes have been theoretically predicted [20], only a few have been synthesized. MXenes are applied in a wide range of fields, including electrocatalysis [21], hydrogen storage [22,23], lithium-ion batteries [24,25], and supercapacitors [26]. MXene is a potential candidate for electrochemical NRRs (e-NRRs) because of its large specific surface area, adjustable structure, and excellent stability [27,28,29].

MXene-based electrocatalysts for the e-NRR can be divided into two categories: pure MXene and MXene-based hybrid electrocatalysts [30]. Pure MXene is a potential candidate for the e-NRR. For example, Azofra et al. [31] found that M_3_C_2_ exhibited good N_2_ capture and activation behavior. However, bare-metal atoms on the surface of M_3_C_2_ are considered active sites [31,32], which tend to bind to functional groups such as oxygen groups; thus, the electrical conductivity is decreased, and the active sites are inactivated. Pure MXene still faces challenges as a catalyst for the e-NRR; therefore, MXene hybrids have been designed. Li et al. [33] loaded nanosized Au particles onto Ti_3_C_2_ nanosheets (Au/Ti_3_C_2_) for the e-NRR. Their research indicated that the hybrid is conducive to N_2_ chemisorption and decreases the activation energy barrier. Au/Ti_3_C_2_ shows excellent catalytic performance. MnO_2_-decorated Ti_3_C_2_T*_x_* (MnO_2_-Ti_3_C_2_T*_x_*) has also been studied as an efficient electrocatalyst for ammonia synthesis under environmental conditions [34]. MnO_2_ and Ti_3_C_2_T*_x_* synergistically promote electrocatalytic activity to achieve superior catalytic activity. In addition, single-atom catalysts (SACs) have been widely studied because of their low cost, superior performance, and full use of metal atoms. Gao et al. [5] studied the reaction pathways and overpotentials of Ti_3_C_2_O_2_-supported TM (Fe, Co, Ru, Rh) SACs. These MXene hybrids, including noble metal–MXene, TM oxide–MXene, and MXene-based SACs, have effectively changed the catalytic performance, providing more possibilities for the screening of new efficient and stable catalysts.

In this study, a 2D MXene, Ti_3_C_2_O_2_, was modified with nonmetals (N, F, P, S, and Cl) and adsorbed TM (Fe atom, Fe/Ti_3_C_2_O_2−*x*_) to study the catalytic performance of the e-NRR. Gibbs free energy (Δ*G*) was used to analyze the reaction pathway and limit the potential of each catalyst, and the main potential-limiting steps of the reaction were determined as *N_2_ + H → *NHH and *NH_3_ → NH_3_.

## 2. Computational Methods

Density functional theory (DFT) calculations were performed using the Vienna ab initio simulation package v. 5.4.4. (University of Vienna, Vienna, Austria) [35,36]. The generalized gradient approximation with Perdew–Burke–Ernzerhof was used as an exchange-correlation function [37]. The projector-augmented wave method was adopted to describe the effect of the core electrons on the valence electron density [38]. The cut-off energy was set to 600 eV. The convergence criteria for the energy and force were 10^−5^ eV and 10^−2^ eV/Å, respectively. The thickness of the vacuum layer was more than 20 Å to avoid interactions in the *z*-direction, and the *x*-and *y*-directions were set as periodic boundary conditions. A 3 × 3 × 1 supercell was used for all the structures. For geometric optimization, the Brillouin zones were sampled with 4 × 4 × 1 Monkhorst–Pack meshes [39], and DFT-D3 was used to accurately describe Van der Waals interactions [40]. Charge transfer was computed by Bader charge population analysis [41,42] and the electron localization function (ELF) was analyzed using the VESTA code [43].

The substitution energies (Δ*E*_sub_) of doping different nonmetallic elements (N/F/P/S/Cl) on the surface of Ti_3_C_2_O_2_ can be expressed as
(1)$$\Delta E_{\mathrm{sub}}=E_{\mathrm{NM}-{\mathrm{Ti}}_{3}C_{2}O_{2-x}}-E_{{\mathrm{Ti}}_{3}C_{2}O_{2}}+E_{O}-E_{\mathrm{NM}}$$
where *E*_O_ and *E*_NM_ represent the energies of a single O atom and nonmetallic elements (N, F, P, S, Cl), respectively, and were calculated using H_2_ [44], H_2_O [45], NH_3_ [46], HF [47], H_3_PO_4_ [48], H_2_S [49], and HCl [50] from the Open Quantum Materials Database (OQMD) [51,52].

The adsorption energy (Δ*E*_ads_) of Fe anchored on NM-Ti_3_C_2_O_2−*x*_ (NM represents the surface nonmetals, O, N, F, P, S, and Cl) was calculated using the following formula:(2)$$\Delta E_{\mathrm{ads}}=E_{\mathrm{Fe}/\mathrm{NM}-{\mathrm{Ti}}_{3}C_{2}O_{2-x}}-E_{\mathrm{NM}-{\mathrm{Ti}}_{3}C_{2}O_{2-x}}-E_{\mathrm{Fe}}$$

Δ*G* was calculated as described by Nørskov et al. [53]. Under standard reaction conditions, the chemical potential of a proton and electron pair (μ[H^+^ + e^−^]) is equal to half that of gaseous hydrogen (μ[H_2_]). Δ*G* was calculated using the following formula:(3)$$\Delta G=\Delta E_{\mathrm{DFT}}+\Delta ZPE-T\Delta S-neU+\Delta G_{\mathrm{pH}}$$
where Δ*E* is the potential energy change calculated by DFT, Δ*ZPE* is the zero-point energy correction, and it is calculated by calculating the frequency of the adsorbed species. *T*Δ*S* is the entropy correction, which is usually available from some database, where *T* = 298 K; Δ*G*_pH_ and *neU* are the contributions from the pH and electrode potential (*U*), respectively; *n* is the number of electrons transferred; *U* is the applied bias. Δ*G*_pH_ is defined as
(4)$$\Delta G_{\mathrm{pH}}=-k_{B}T\ln\left[H^{+}\right]=\mathrm{pH}\times K_{B}T\ln10$$
where *k*_B_ is Boltzmann’s constant. For all the calculations, the pH was set to zero. The Δ*E*_ads_ values of different adsorbates were calculated as follows:(5)$$\Delta E_{\mathrm{ads}}=E_{\mathrm{cat}–\mathrm{mol}}-E_{\mathrm{cat}}-E_{\mathrm{mol}}$$
where Δ*E*_cat–mol_ is the energy of the entire adsorption structure, *E*_cat_ is the energy of the catalyst, and *E*_mol_ is the energy of the adsorbate molecules such as N_2_ and N*_x_*H*_y_*.

## 3. Results and Discussion

### 3.1. Geometric Structure

Bare Ti_3_C_2_ is a hexagonal lattice with P$\bar{3}$*m*1 group symmetry, five atomic layers of Ti–C–Ti–C–Ti, two exposed Ti layers, and an experimental lattice constant of 3.057 Å [54]. After structural optimization, *a* = *b* = 3.020 Å, which was in good agreement with the experimental values. Bare MXenes are unstable under relevant NRR operating conditions [55], and they are always functionalized by electronegative functional groups [56], as they are chemically exfoliated from the bulk MAX phase by HF [17,57]. O-terminated Ti_3_C_2_ was used for further experiments. There are different possibilities for the adsorption of O on Ti_3_C_2_. According to previous studies [5], the most stable structure is O adsorbed at the hollow sites of the contralateral surface Ti atoms, as shown in Figure 1a,b. Nonmetallic elements (N/F/P/S/Cl) were used to modify the Ti_3_C_2_O_2_ surface. Δ*E*_sub_ indicates the stability of a surface before and after doping with nonmetallic elements. The Δ*E*_sub_ values for N, F, P, S, and Cl were 1.79, −1.04, 0.81, −0.27, and −1.01 eV, respectively. The structure became more stable after doping with F, S, and Cl when Δ*E*_sub_ < 0 and became more unstable after doping with N and P when Δ*E*_sub_ > 0. Among these doping situations, doping with F had the best stability, compared with doping with other nonmetallic elements.

Pure Ti_3_C_2_O_2_ and Ti_3_C_2_O_2_ modified with nonmetallic elements (Appendix A) were used to support single Fe atoms. Two different hollow sites (H1 and H2) and an O-top site on the surface were considered, as shown in Figure 1a. The O-top was unstable, and the *E*_ads_ values of Fe adsorbed on H1 and H2 are listed in Table 1. Except for the F-doped structures, the Fe atoms preferred to adsorb on the H1 site, as the *E*_ads_ was smaller. Notably, in the F-doped structure, the Fe atom was adsorbed on the next-nearest H1 site (Figure 1e). As shown in Table 1, the doping of N, F, P, and S facilitates the adsorption of Fe, while it is more difficult for Fe to adsorb on the Cl-doped structure. Figure 1c–h show the most stable adsorption positions for the different catalysts.

### 3.2. N_2_ Adsorption

Based on the Fe/NM-Ti_3_C_2_O_2−*x*_ structure, N_2_ adsorption was calculated using *E*_ads_. There are two different positions for N_2_ adsorption, and advanced research has shown that N_2_ adsorption is closer end to end than side to side [5]. Figure 2a–f show the most stable structure of N_2_ adsorbed on different catalysts from end to end, and Figure 2g–l show the ELF of these structures. *E*_ads_ ranged from −0.55 eV to −0.92 eV, which indicates that the N_2_ adsorption has strong spontaneity, and the absolute value of *E*_ads_ from small to large was in the order: Fe/P-Ti_3_C_2_O_2−*x*_ < Fe/S-Ti_3_C_2_O_2−*x*_ < Fe/N-Ti_3_C_2_O_2−*x*_ < Fe/F-Ti_3_C_2_O_2−*x*_ < Fe/Cl-Ti_3_C_2_O_2−*x*_ < Fe/Ti_3_C_2_O_2_ (Table 1). After N_2_ adsorption, the N≡N bond lengths in Fe/Ti_3_C_2_O_2_, Fe/N-Ti_3_C_2_O_2−*x*_, Fe/F-Ti_3_C_2_O_2−*x*_, Fe/P-Ti_3_C_2_O_2−*x*_, Fe/S-Ti_3_C_2_O_2−*x*_, and Fe/Cl-Ti_3_C_2_O_2−*x*_ are 1.128, 1.125, 1.129, 1.123, 1.126, and 1.130 Å, respectively. Compared with the N≡N bond length in the gas phase (1.11 Å), all of them became longer. The calculation of charge transfer is shown in Table 1. The results show that N_2_ gains electrons in all these catalysts and the translated charges increase with an increase in the number of valence electrons from N to O or from P to S and Cl in the same period. However, doping with F did not obey this rule, which may be due to the special adsorption site of Fe. Fe was adsorbed on the first nearest H1 site and followed the trend from N to O and F. These findings were consistent with those of Wang et al. [58]. A strong positive correlation exists between the electron gains of N_2_ and the change in bond length: N_2_ on Fe/Cl-Ti_3_C_2_O_2−*x*_ gained the most electrons and had the largest increase in bond length relative to the gas phase, whereas N_2_ on Fe/P-Ti_3_C_2_O_2−*x*_ gained the least electrons and had the smallest increment in bond length relative to the gas phase.

The partial density of states of N_2_ adsorbed on Fe/Ti_3_C_2_O_2_ or Fe/NM-Ti_3_C_2_O_2−*x*_ (Figure 3) shows spin-up and spin-down of the d orbital of the Fe atom and the p orbital of the N atom. At the Fermi level, almost no spin-up was observed, whereas the spin-down was more obvious, and the d orbital of Fe effectively overlapped with the P orbital of N near the Fermi level. The electrons in the occupied d orbital of Fe/NM-Ti_3_C_2_O_2−*x*_ transferred to the antibonding orbitals of N_2_, as shown in Table 1, and the adsorbed N_2_ on different catalysts gained electrons from 0.13 e to 0.21 e, thus lowering the bond energy of N_2_.

### 3.3. N_2_ Reduction Mechanism

The overall e-NRR reaction on the cathode is
(6)$$N_{2}\left(g\right)+6\left(H^{+}+e^{-}\right)\to 2{\mathrm{NH}}_{3}\left(g\right)$$
and the anode reactions provide protons and electrons. Liu et al. [59] summarized the mechanism of the e-NRR. The e-NRR is divided into dissociation and association mechanisms by different hydrogenation (protonation and reduction) sequences and the breaking of the N≡N triple bond. In the dissociation mechanism, the N≡N bond is broken during the adsorption process (* denotes the adsorption site).
(7)$$2*+N_{2}\to 2*N$$

Then, two separated N atoms on the surface of the catalysts receive protons and electrons, and ammonia is formed in the last hydrogenation step:(8)$$*N+H^{+}+e^{-}\to *\mathrm{NH}$$
(9)$$*\mathrm{NH}+H^{+}+e^{-}\to *{\mathrm{NH}}_{2}$$
(10)$$*{\mathrm{NH}}_{2}+H^{+}+e^{-}\to *{\mathrm{NH}}_{3}$$
(11)$$*{\mathrm{NH}}_{3}\to {\mathrm{NH}}_{3}$$

In the association mechanism, the N≡N bond breaks at a certain hydrogenation step. According to the hydrogenation sequence, it can be further classified into distal, alternating, and enzymatic pathways. The hydrogenation step in the enzymatic pathway is similar to that in the alternating pathway; the difference is that N_2_ adsorbs side to side in the enzymatic pathway, but ends in the distal and alternating pathways. For the distal and alternating pathways, the first two steps are
(12)$$*+N_{2}\to *N_{2}$$
(13)$$*N_{2}+H^{+}+e^{-}\to *N_{2}H$$

In the distal pathway, the N atom moves away from the catalytically gained protons and electrons, releasing the first NH_3_ molecule, as follows:(14)$$*N_{2}H+H^{+}+e^{-}\to *{\mathrm{NNH}}_{2}$$
(15)$$*{\mathrm{NNH}}_{2}+H^{+}+e^{-}\to *N+{\mathrm{NH}}_{3}$$

Hydrogenation then occurs on the remaining N atom and releases the second NH_3_ molecule according to Reactions (8)–(11). In the alternating pathway, hydrogenation occurs on two newton atoms alternatively, and NH_3_ is formed until the N≡N bond is completely broken.
(16)$$*N_{2}H+H^{+}+e^{-}\to *\mathrm{NHNH}$$
(17)$$*\mathrm{NHNH}+H^{+}+e^{-}\to *{\mathrm{NHNH}}_{2}$$
(18)$$*{\mathrm{NHNH}}_{2}+H^{+}+e^{-}\to *{\mathrm{NH}}_{2}{\mathrm{NH}}_{2}$$
(19)$$*{\mathrm{NH}}_{2}{\mathrm{NH}}_{2}+H^{+}+e^{-}\to *{\mathrm{NH}}_{2}+{\mathrm{NH}}_{3}$$

After the first NH_3_ is released, the remaining *NH_2_ obtains protons and electrons and releases the second ammonia according to Reactions (10) and (11). Figure 4 shows the other mixed pathways that follow neither the distal nor alternating pathways but a combination of two paths. Optimized structures of all the possible elementary steps in NRR is showed in Appendix A. 

The Δ*G* values calculated by DFT calculations considered all correction terms, including the zero-point energy, temperature, and entropy corrections. Table 2 illustrates the *E*_ZPE_ and entropy corrections (*TS*) of different reaction intermediates on Fe/Ti_3_C_2_O_2_ using the *TS* values obtained from the National Institute of Standards and Technology [60] at *T* = 298 K. The catalyst as a substrate is immobilized, although the surface is different, we compared the zero-point energy with the study of Ling [61]; the difference is marginal, as N_2_ reduction also occurred on the transition metal atoms in Ling’s research, and only the *E*_ZPE_ of NH_3_ was significantly different. NH_3_ is a gas phase, not an adsorbent, so other research was also compared [5]. The calculated *E*_ZPE_ and *TS* of H_2_ are 0.27 and 0.4 eV [60], respectively.

As shown in Figure 5a–f, for all structures, the first protonation was likely to generate *NNH species; the Δ*G* values for Fe/Ti_3_C_2_O_2_, Fe/N-Ti_3_C_2_O_2−*x*_, Fe/F-Ti_3_C_2_O_2−*x*_, Fe/P-Ti_3_C_2_O_2−*x*_, Fe/S-Ti_3_C_2_O_2−*x*_, and Fe/Cl-Ti_3_C_2_O_2−*x*_ increased to 0.90, 1.04, 0.85, 0.99, 0.88, and 1.01 eV, respectively. The second step is more likely to form *NNH_2_ instead of the *NHNH species in the alternate path, as the energy requirements are higher, and the increments in Δ*G* for Fe/Ti_3_C_2_O_2_, Fe/N-Ti_3_C_2_O_2−*x*_, Fe/F-Ti_3_C_2_O_2−*x*_, Fe/P-Ti_3_C_2_O_2−*x*_, Fe/S-Ti_3_C_2_O_2−*x*_, and Fe/Cl-Ti_3_C_2_O_2−*x*_ were 0.1, 0.06, 0.12, −0.05, 0.12, and 0.07 eV to form *NNH_2_, respectively. In the subsequent hydrogenation steps, the intermediate configuration in the alternating pathway was easier to form than the first NH_3_ molecule desorption in the distal pathway. The first NH_3_ is not desorbed until the fifth proton is added, and adsorptive *NH_3_ is formed when the sixth proton is added. The reaction *NNH_2_ → *NHNH_2_ → *NH_2_NH_2_ → *NH_2_ → *NH_3_ is exothermic, and larger energy input is required until the adsorptive *NH_3_ is desorbed to form the second NH_3_ molecule. The Δ*G* values of Fe/Ti_3_C_2_O_2_, Fe/N-Ti_3_C_2_O_2−*x*_, Fe/F-Ti_3_C_2_O_2−*x*_, Fe/P-Ti_3_C_2_O_2−*x*_, Fe/S-Ti_3_C_2_O_2−*x*_, and Fe/Cl-Ti_3_C_2_O_2−*x*_ were 1.95, 1.11, 0.97, 1.07, 1.09, 0.99 eV, respectively. However, it was reported that the use of an acidic electrolyte can promote NH_3_ desorption, as the protonation of adsorbed NH_3_ to form NH4^+^ can easily proceed [62,63], so the actual energy barrier is even smaller. For all these structures, the two potential limiting steps were the first hydrogenation of N_2_ to form the *NNH species and the last process of NH_3_ desorption to form the second NH_3_ molecule. Compared with the original structure, nonmetallic doping was beneficial for the desorption of the last NH_3_ molecule, but only the doping of F and S was beneficial for the formation of *NNH and NH_3_.

Figure 6 shows the most possible reaction pathway for different catalysts. All these structures are likely to follow the mixed pathway: N_2_ → *N_2_ → *NNH → *NNH_2_ → *NHNH_2_ → *NH_2_NH_2_ → *NH_2_ → *NH_3_ → NH_3_. In addition, the doping of nonmetals has a remarkable effect on NRR. For N_2_ adsorption, *E*_ads_ is reduced, compared with the nondoped structure, which may be the reason why NH_3_ desorption is easier in the last step. In the hydrogenation process, the doping of different nonmetals also makes each step of the hydrogenation easier or harder. The doping of N, P, and Cl makes it difficult for *N_2_ to form *NNH, whereas F and S facilitate the formation of *NNH from *N_2_. From *NNH to *NNH_2_, only the doping of P shows an obvious impact and makes the transformation occur spontaneously. In comparison, the other doped nonmetals do not show a great effect. The doping of nonmetal also does not have much influence on *NNH_2_ → *NHNH_2_ → *NH_2_NH_2_ → *NH_2_ → *NH_3_, as these reactions are exothermic for all structures. Considering the stability of nonmetal doping, the best catalysts may be Fe/F-Ti_3_C_2_O_2−*x*_ and Fe/S-Ti_3_C_2_O_2−*x*_.

## 4. Conclusions

The reaction pathway of the TM atom, Fe, adsorbed on pure Ti_3_C_2_O_2_ and surface nonmetal (N/F/P/S/Cl)-doped Ti_3_C_2_O_2_ as the N_2_ reduction reaction catalyst was calculated using DFT. The main limiting steps of the reaction are *N_2_ + H → *NNH and *NH_3_ → NH_3_, and the limiting potentials of the two steps can reach 0.85–1.01 and 0.97–1.95 eV, respectively. Compared with pure Ti_3_C_2_O_2_, nonmetal doping has an impact on catalytic performance. The doped nonmetal (N/F/P/S/Cl) reduces the energy barrier to form NH_3_ in the last step, and only the doping of F and S is beneficial to the formation of *NNH in the first step and the desorption of *NH_3_ in the last step. Therefore, the materials doped with F and S are considered better candidate materials for NRR among the tested catalysts. Our research demonstrates a feasible way to search for new NRR catalysts by modifying the surface of MXenes and loading TM atoms as new catalysts.

## Figures and Tables

**Figure 1 nanomaterials-12-01081-f001:**
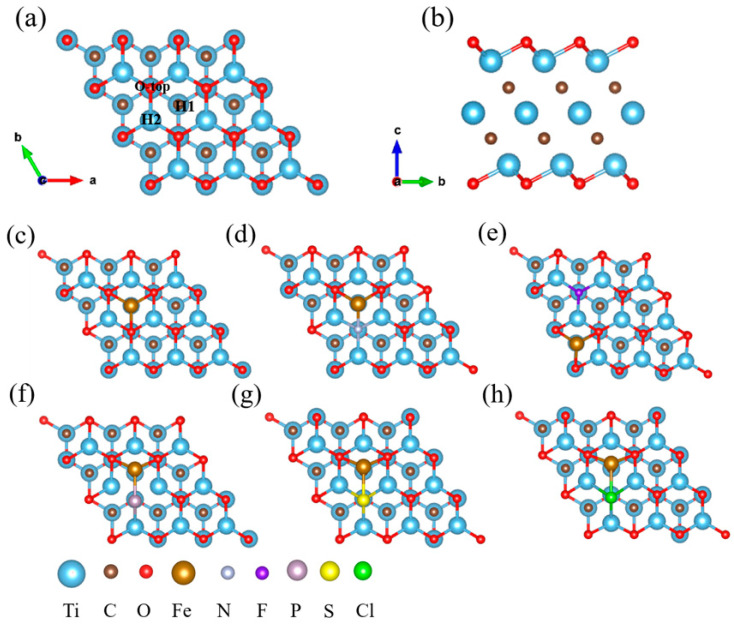
(**a**) Top view and different adsorption sites on Ti_3_C_2_O_2_ and (**b**) side view of Ti_3_C_2_O_2_. The most stable structure of Fe adsorbed on (**c**) Ti_3_C_2_O_2_, (**d**) N-doped Ti_3_C_2_O_2_, (**e**) F-doped Ti_3_C_2_O_2_, (**f**) P-doped Ti_3_C_2_O_2_, (**g**) S-doped Ti_3_C_2_O_2_, and (**h**) Cl-doped Ti_3_C_2_O_2_.

**Figure 2 nanomaterials-12-01081-f002:**
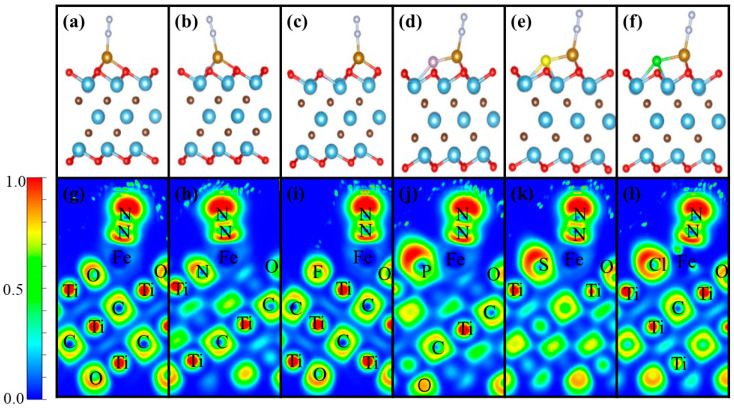
Most stable structures of N_2_ adsorbed on (**a**) Fe/Ti_3_C_2_O_2_, (**b**) Fe/N-Ti_3_C_2_O_2−*x*_, (**c**) Fe/F-Ti_3_C_2_O_2−*x*_, (**d**) Fe/P-Ti_3_C_2_O_2−*x*_, (**e**) Fe/S-Ti_3_C_2_O_2−*x*_, and (**f**) Fe/Cl-Ti_3_C_2_O_2−*x*_ and ELFs of N_2_ adsorbed on (**g**) Fe/Ti_3_C_2_O_2_, (**h**) Fe/N-Ti_3_C_2_O_2−*x*_, (**i**) Fe/F-Ti_3_C_2_O_2−*x*_, (**j**) Fe/P-Ti_3_C_2_O_2−*x*_, (**k**) Fe/S-Ti_3_C_2_O_2−*x*_, and (**l**) Fe/Cl-Ti_3_C_2_O_2−*x*_.

**Figure 3 nanomaterials-12-01081-f003:**
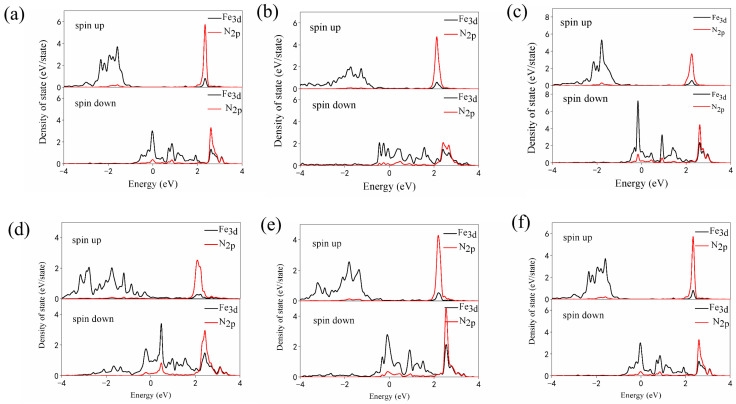
Partial density of states of N_2_ adsorbed on (**a**) Fe/Ti_3_C_2_O_2_, (**b**) Fe/N-Ti_3_C_2_O_2−*x*_, (**c**) Fe/F-Ti_3_C_2_O_2−*x*_, (**d**) Fe/P-Ti_3_C_2_O_2−*x*_, (**e**) Fe/S-Ti_3_C_2_O_2−*x*_, and (**f**) Fe/Cl-Ti_3_C_2_O_2−*x*_.

**Figure 4 nanomaterials-12-01081-f004:**
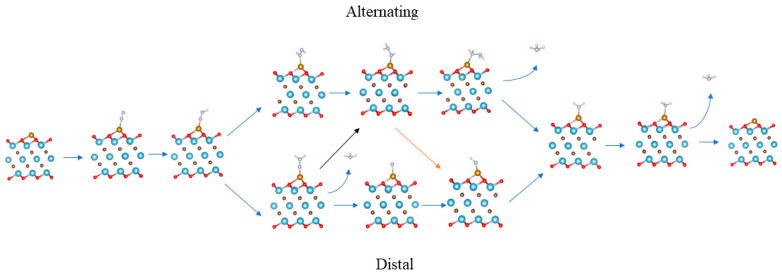
Possible pathway and reaction intermediates for NRR with the associated mechanism. Dark brown, blue, red, brown, light blue, and light pink represent C, Ti, O, Fe, N, and H, respectively.

**Figure 5 nanomaterials-12-01081-f005:**
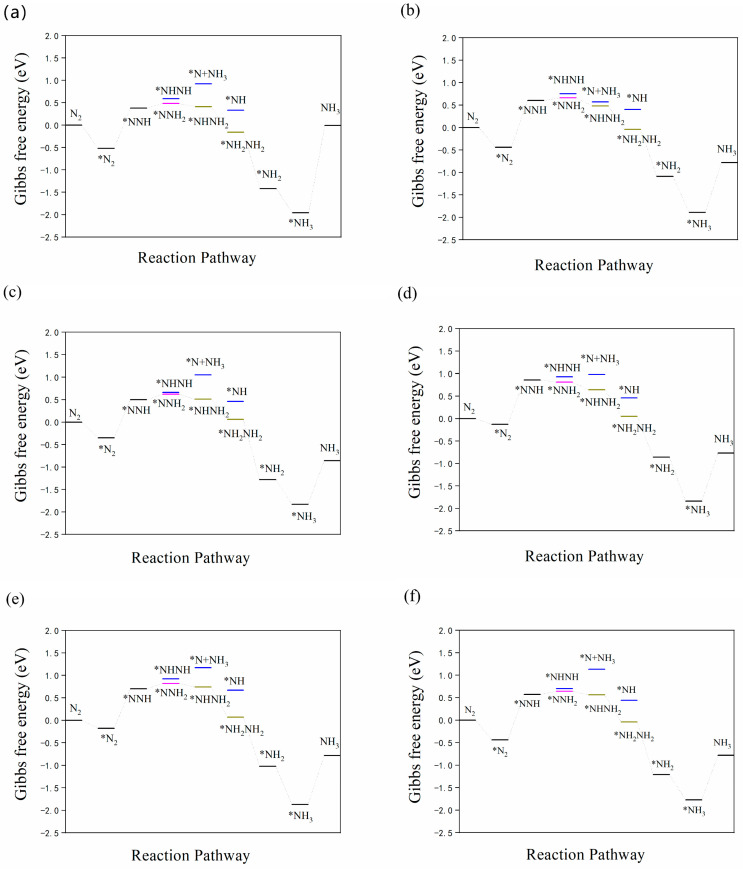
Gibbs free energy diagrams of (**a**) Fe/Ti_3_C_2_O_2_, (**b**) Fe/N-Ti_3_C_2_O_2−*x*_, (**c**) Fe/F-Ti_3_C_2_O_2−*x*_, (**d**) Fe/P-Ti_3_C_2_O_2−*x*_, (**e**) Fe/S-Ti_3_C_2_O_2−*x*_, and (**f**) Fe/Cl-Ti_3_C_2_O_2−*x*_.

**Figure 6 nanomaterials-12-01081-f006:**
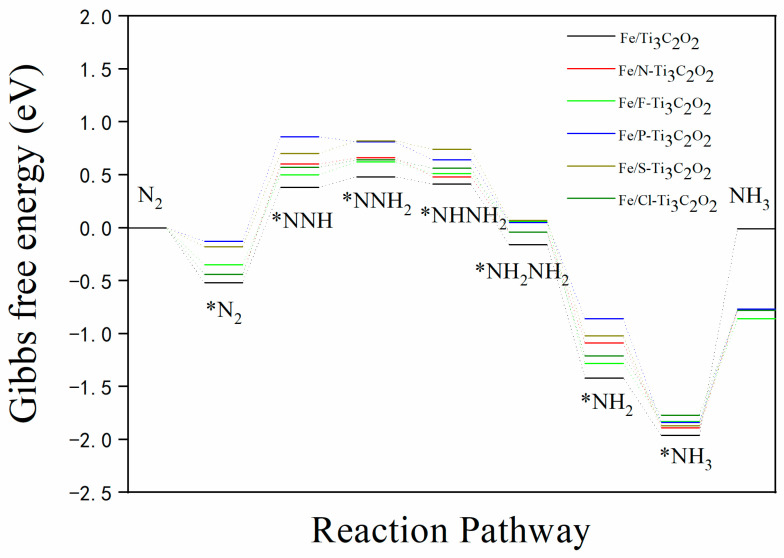
Nitrogen reduction reaction pathways for all structures.

**Table 1 nanomaterials-12-01081-t001:** Adsorption energies of Fe adsorbed on different sites and N_2_ adsorbed on different catalysts, the charge on N_2_, and the charge transferred after N_2_ adsorption.

Species	*E*_ads_ of Fe (eV)		*E*_ads_ of N_2_ (eV)	Charge Transferred on N_2_ (e)
	H1	H2		
Fe/Ti_3_C_2_O_2_	−3.57	−3.30	−0.92	0.19
Fe/N-Ti_3_C_2_O_2−*x*_	−4.32	−3.90	−0.77	0.15
Fe/F-Ti_3_C_2_O_2−*x*_	−3.61	−3.60	−0.78	0.18
Fe/P-Ti_3_C_2_O_2−*x*_	−5.12	−4.68	−0.55	0.13
Fe/S-Ti_3_C_2_O_2−*x*_	−4.33	−4.02	−0.59	0.16
Fe/Cl-Ti_3_C_2_O_2−*x*_	−3.39	−3.11	−0.85	0.21

**Table 2 nanomaterials-12-01081-t002:** *E*_ZPE_ and *TS* of different reaction intermediates on Fe/Ti_3_C_2_O_2_, *T* = 298 K.

Adsorption Species	*E*_ZPE_ (eV)	*E*′_ZPE_ (eV)	*E*_ZPE_ Difference (eV)	*TS* [60] (eV)
N_2_	0.15	0.15 [61]	0	0.59
*N≡N	0.19	0.20 [61]	0.01	0.23
*N=NH	0.47	0.49 [61]	0.02	0.20
*N−NH_2_	0.78	0.82 [61]	0.04	0.25
*N	0.09	0.08 [61]	0.01	0.06
*NH	0.31	0.35 [61]	0.04	0.14
*NH_2_	0.63	0.65 [61]	0.02	0.18
*NH_3_	1.00	1.02 [61]	0.02	0.23
*NH=NH	0.81	0.80 [61]	0.01	0.25
*NH−NH_2_	1.11	1.13 [61]	0.02	0.31
*NH_2_−NH_2_	1.50	1.49 [61]	0.01	0.27
NH_3_	0.92	0.96 [5]	0.04	0.60

## Data Availability

The datasets generated during and/or analyzed during the current study are available from the corresponding author.

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
