# Peer review of "A Theoretical Study of Fe Adsorbed on Pure and Nonmetal (N, F, P, S, Cl)-Doped Ti3C2O2 for Electrocatalytic Nitrogen Reduction"

_nanomaterials, 2022, doi:10.3390/nano12071081_

Round 1

Reviewer 1 Report

Report on the manuscript “A Theoretical Study of Fe Adsorbed on Pure and Nonmetal (N, 2 F, P, S, Cl)-Doped Ti3C2O2 for Electrocatalytic Nitrogen 3 Reduction”, H. Luo et al.

Ref. Nanomaterials_1643925.

General comments:

The manuscript presents a theoretical study dealing with the possibility of using transition metal (TM)/MXene (TM = Fe, MXene = Ti3C2O2 or Ti3C2O2−x doped with N/F/P/S/Cl) as a catalyst for the nitrogen reduction reaction (NRR) using the density functional theory. Overall, the manuscript presents an interesting study, but several aspects need clarification as detailed below, so that minor revision is recommended based on the following considerations.

General remarks:

  1. I) The authors discuss the different pathways for the NRR resulting in the formation of NH3 gas from a N2-containing aqueous solution. However, for all the calculations, the pH was set to zero (line 111). Under these conditions, the electrochemical reduction of N2 will yield ammonium ion (NH4+) rather than ammonia. Some clarification is needed.

Minor remarks:

1) The meaning of neU in Eq. (3) should be clarified.

2) In Eq. (4), thermochemical activity rather than concentration should be used although, as customary, pH’s were referred to concentrations.

3) The sentence in line 188: “and the anode reactions served as protons and electrons” has an unclear meaning.

Author Response

Response to Reviewer 1 Comments

Dear Editors and Reviewers:

Thank you very much for your letter and for the reviewers’ comments on our manuscript entitled “A Theoretical Study of Fe Adsorbed on Pure and Nonmetal (N, F, P, S, Cl)-Doped Ti3C2O2 for Electrocatalytic Nitrogen Reduction” (ID: 1643925). Those comments are valuable and very helpful for revising and improving our manuscript. We have considered these comments carefully and revised our manuscript accordingly. Changes are marked in red in the revised manuscript. In the following, we address the reviewers’ comments point by point.

Point 1: The authors discuss the different pathways for the NRR resulting in the formation of NH3 gas from a N2-containing aqueous solution. However, for all the calculations, the pH was set to zero (line 111). Under these conditions, the electrochemical reduction of N2 will yield ammonium ion (NH4+) rather than ammonia. Some clarification is needed.

Response 1: Thanks for your valuable comments. As the protonation of adsorbed NH3 to form NH4+ can easily proceed, the use of an acidic electrolyte can promote NH3 desorption [1,2], It has been added in the manuscript (page 8, line 248).

Point 2: The meaning of neU in Eq. (3) should be clarified.

Response 2: Thanks for your suggestion. neU in Eq. (3) is the contributions from electrode potential (U), and n is the number of electrons transferred, U is the applied bias. It has been clarified in the manuscript (page 3, line 107).

Point 3: In Eq. (4), thermochemical activity rather than concentration should be used although, as customary, pH’s were referred to concentrations.

Response 3: Thanks for your comments. In related studies [3], the PH is usually set to zero, and this term loses its role in calculating energy.

Point 4: The sentence in line 188: “and the anode reactions served as protons and electrons” has an unclear meaning.

Response 4: Thanks for your carefully review. There is a clerical error here, which has been corrected as “and the anode reactions provides protons and electrons” (line 188).

References

  1. J. Chun, V. Apaja, A. Clayborne, K. Honkala, J. Greeley, Atomistic Insights into Nitrogen-Cycle Electrochemistry: A Combined DFT and Kinetic Monte Carlo Analysis of NO Electrochemical Reduction on Pt(100), ACS Catal. 7 (2017) 3869–3882. https://doi.org/10.1021/acscatal.7b00547.
  2. Clayborne, H.-J. Chun, R.B. Rankin, J. Greeley, Elucidation of Pathways for NO Electroreduction on Pt(111) from First Principles, Angew. Chemie. 127 (2015) 8373–8376. https://doi.org/10.1002/ange.201502104.
  3. Gao, H. Zhuo, Y. Cao, X. Sun, G. Zhuang, S. Deng, X. Zhong, Z. Wei, J. Wang, A theoretical study of electrocatalytic ammonia synthesis on single metal atom/MXene, Cuihua Xuebao/Chinese J. Catal. 40 (2019) 152–159. https://doi.org/10.1016/S1872-2067(18)63197-3..

Reviewer 2 Report

The manuscript entitled “A Theoretical Study of Fe Adsorbed on Pure and Nonmetal (N, F, P, S, Cl)-Doped Ti3C2O2 for Electrocatalytic Nitrogen Reduction” by Heng Luo, Xiaoxu Wang, Chubin Wan, Lu Xie, Minhui Song, Ping Qian deals with reactions of N2 and H2 on the surface of modified Ti3C2O2. The study might be interesting for the readership of the Nanomaterials journal. However, I miss some calculation details and I do not understand several calculation aspects in the current form of the manuscript.

1) In my understanding, the authors use a very small cell in the z-coordinate as shown in Figure 1b. Do they keep some atoms fixed when optimizing structure after introduction of the vacuum layer? The authors should show that the number of layers does not influence their results considerably, at least through analysis of N2 adsorption on a thicker structure. At the moment, it seems that they simulate more a “nanolayer” than a surface.

2) Table 1: Charge on N2 is given as, e.g., 10.19. The authors probably mean the number of electrons on N2. This column can be left out as the “transferred” column includes all information that is needed.

3) Table 2: I do not understand how the values were obtained. a) Is the zero-point energy calculated just for stretch frequencies of the adsorbed species? Are bending modes included or somehow projected from the overall frequencies calculation? b) The second column is probably taken from the SI of Ref. 59, but some values are different compared to Table S1 of the respective publication. Also, for which surface were the values calculated, can one directly compare them to the current results? c) What does the “relative error” mean, what is compared here? d) The authors claim that the T*S values were obtained from the National Institute of Standards and Technology. Do the respective tables really include T*S correction for adsorbed NxHy species? Why are these contribution not calculated directly from the data the authors obtained? e) On Line 233, the authors state “The calculated EZPE and TS of N2 are 0.27 and 0.4 eV.[58]” How does this correspond to the values in the table?

4) All optimized structures should be included in the Supporting Information file.

Typos:

Line 27: Hubble-Bosch -> Haber-Bosch

Line 77: “*N2 + H -> *NHH” should read “*N2 + H -> *NNH”

Line 81: “function” -> “functional”

Line 117: “Geometric structure” -> “Structure”

Line 215: “newton” -> “nitrogen” (?)

Author Response

Response to Reviewer 2 Comments

Dear Editors and Reviewers:

Thank you very much for your letter and for the reviewers’ comments on our manuscript entitled “A Theoretical Study of Fe Adsorbed on Pure and Nonmetal (N, F, P, S, Cl)-Doped Ti3C2O2 for Electrocatalytic Nitrogen Reduction” (ID: 1643925). Those comments are valuable and very helpful for revising and improving our manuscript. We have considered these comments carefully and revised our manuscript accordingly. Changes are marked in red in the revised manuscript. In the following, we address the reviewers’ comments point by point.

Point 1: In my understanding, the authors use a very small cell in the z-coordinate as shown in Figure 1b. Do they keep some atoms fixed when optimizing structure after introduction of the vacuum layer? The authors should show that the number of layers does not influence their results considerably, at least through analysis of N2 adsorption on a thicker structure. At the moment, it seems that they simulate more a “nanolayer” than a surface.

Response 1: Thanks for your valuable comments. We did not fixed atoms, but fixed vacuum layer. As a member of MXenes, Ti3C2O2 is a two-dimensional nanomaterial, Ti3C2 have five atomic layers of Ti–C–Ti–C–Ti, it becomes another MXenes material with the layers changed. We optimize the overall structure of Ti3C2O2, and when the vacuum layer is large enough [1,2], in fact, we set 27 Å to avoid the interaction between layers.

Point 2: Table 1: Charge on N2 is given as, e.g., 10.19. The authors probably mean the number of electrons on N2. This column can be left out as the “transferred” column includes all information that is needed.

Response 2: Thanks for your suggestion. We deleted the column of “Charge on N2”, and Table 1 has been corrected in the manuscript (page 4, line 148).

Point 3: Table 2: I do not understand how the values were obtained. a) Is the zero-point energy calculated just for stretch frequencies of the adsorbed species? Are bending modes included or somehow projected from the overall frequencies calculation? b) The second column is probably taken from the SI of Ref. 59, but some values are different compared to Table S1 of the respective publication. Also, for which surface were the values calculated, can one directly compare them to the current results? c) What does the “relative error” mean, what is compared here? d) The authors claim that the T*S values were obtained from the National Institute of Standards and Technology. Do the respective tables really include T*S correction for adsorbed NxHy species? Why are these contribution not calculated directly from the data the authors obtained? e) On Line 233, the authors state “The calculated EZPE and TS of N2 are 0.27 and 0.4 eV.[58]” How does this correspond to the values in the table?

Response 3: Thanks for asking these questions. a) The zero-point energy is calculated by calculating the frequency of the adsorbed species. We use VASP calculated the vibration mode and frequency of the adsorbed species, and obtained the zero-point energy through mathematical sum calculation. b) The only difference in the second column is the zero-point energy of NH3, which is significantly different from ling's work [3] (Ref. 59), NH3 is a gas phase, not an adsorbent, so we also compared other work (Ref. 5). Because the catalyst that acts as the base is fixed, only the adsorbed species are considered, although the surface is different, it is reasonable to make a direct comparison. In addition, Ling's work is also carried out on transition metal atoms. c) The “relative error” is the absolute value of the difference between the first column and the second column, some values have errors and have been modified in the Manuscript (page 7, line 232). As mentioned earlier, in Ling's work, N2 reduction also occurs on transition metal atoms. Therefore, the comparison here is to increase the credibility of zero-point energy calculation. d) VASP does not do very well in entropy calculation. In many related studies, databases are usually cited, such as CRC Handbook of chemistry and physics and National Institute of standards and Technology, they only provide T*S values, but not including the correction, through these values to calculate the correction. e) There is a clerical error here, which has been corrected as “The calculated EZPE and TS of H2 are 0.27 and 0.4 eV.[58]” (page 7, line 231). The zero-point energy and T*S of H2 are not listed in Table 2 and are described separately here.

Point 4: All optimized structures should be included in the Supporting Information file.

Response 4: Thanks for your carefully review. We have put all the optimized structures into the supporting information

References

  1. J.H. Montoya, C. Tsai, A. Vojvodic, J.K. Nørskov, The challenge of electrochemical ammonia synthesis: A new perspective on the role of nitrogen scaling relations, ChemSusChem. 8 (2015) 2180–2186. https://doi.org/10.1002/cssc.201500322.
  2. Y. Gao, H. Zhuo, Y. Cao, X. Sun, G. Zhuang, S. Deng, X. Zhong, Z. Wei, J. Wang, A theoretical study of electrocatalytic ammonia synthesis on single metal atom/MXene, Cuihua Xuebao/Chinese J. Catal. 40 (2019) 152–159. https://doi.org/10.1016/S1872-2067(18)63197-3.
  3. 3. Ling, Y. Ouyang, Q. Li, X. Bai, X. Mao, A. Du, J. Wang, A General Two-Step Strategy–Based High-Throughput Screening of Single Atom Catalysts for Nitrogen Fixation, Small Methods. 3 (2019) 1–8. https://doi.org/10.1002/smtd.201800376.

Round 2

Reviewer 2 Report

I would like to ask the authors to include some of their answers to my point 3) into the Methods section (otherwise, these points are not clear to a reader) and to rename the "relative error" column of Table 2 into, e.g., "ZPE difference" as this is no error.

Otherwise, I recommend to accept the manuscript for publication.

Author Response

Response to Reviewer 2 Comments (Round 2)

Dear Editors and Reviewers:

Thank you very much for your letter and for the reviewers’ comments on our manuscript entitled “A Theoretical Study of Fe Adsorbed on Pure and Nonmetal (N, F, P, S, Cl)-Doped Ti3C2O2 for Electrocatalytic Nitrogen Reduction” (ID: 1643925). Those comments are valuable and very helpful. We have considered these comments carefully and revised our manuscript accordingly. Changes are marked in the revised manuscript. In the following, we address the reviewers’ comments point by point.

Point 1: I would like to ask the authors to include some of their answers to my point 3) into the Methods section (otherwise, these points are not clear to a reader) and to rename the "relative error" column of Table 2 into, e.g., "ZPE difference" as this is no error.

Response 1: Thanks for your valuable comments. We added how to calculate zero-point energy in Manuscript (line 117) and the description of entropy acquisition on line 108 and line 234, and the feasibility of contrast between columns 1 and 2 in Table 2 on line 235 to line 238. In addition, we have renamed the “relative error” column of Table 2 to “EZPE difference” (line 241).